# Patterns and predictors of variability in patient-generated daily pain severity collected via a mobile health smartphone app

Claire L. Little[1], Belay B. Yimer[1], Thomas House[2], William G. Dixon[1,3], David M. Schultz [4,5]*, John McBeth[1,3¤]

**1** Centre for Epidemiology Versus Arthritis, University of Manchester, Manchester, United Kingdom, **2** Department of Mathematics, University of Manchester, Manchester, United Kingdom, **3** NIHR Manchester Musculoskeletal Biomedical Research Unit, Central Manchester University Hospitals NHS Foundation Trust, Manchester, United Kingdom, **4** Centre for Atmospheric Science, Department of Earth and Environmental Sciences, University of Manchester, Manchester, United Kingdom, **5** Centre for Crisis Studies and Mitigation, University of Manchester, Manchester, United Kingdom

¤ Current address: Primary Care Research Centre and Digital Health and Biomedical Engineering Group, University of Southampton, Southampton, United Kingdom
* david.schultz@manchester.ac.uk

## Abstract

Digital-health technologies support the collection of patient-generated health data that is frequent, longitudinal, and collected in participant's own environments. Such high-frequency data could detect patterns of variation in disease and associated symptoms, but characterizing, interpreting, and understanding the reasons for this variability remain open questions. Here, we examine 2070 people living with chronic pain to quantify daily variability in pain severity across seven-day periods and identify factors associated with that variability. Data were collected via a smartphone application from a population-based mobile-health study, Cloudy with a Chance of Pain. Summary statistics and distributions of pain changes on consecutive days were calculated within 13,052 complete weeks of data, which had been assigned to one of four clusters via a previously published k-mediods clustering algorithm: no/low pain, mild pain, moderate pain, and severe pain. Cumulative-probit models were used to identify associations between changes in pain severity and changes in exposure data. Across the four clusters, the no/low-pain cluster had the highest proportion of weeks with no within-week changes (59%) in pain severity compared to the other clusters (48–53%). When pain did change, it changed one unit (out of five) about 20% of the time, but larger changes of two to four units also occurred. Changes in pain severity were associated most strongly with changes in pain interference (i.e., how pain impacts daily activities) and were also associated with changes in fatigue, morning stiffness, mood, and participant well-being. Thus, this study showed that data collected frequently through digital-health technologies can be used to explore variability in symptoms and their associations with other variables. That pain severity was associated with changes in modifiable variables (e.g., fatigue, mood) suggests

**Data availability statement:** The datasets generated and analysed during the current study are not publicly available as data were collected for patient and public involvement activities and consent was obtained for the sharing of anonymous quotes and aggregated data only. The data within the Cloudy with a Chance of Pain study is detailed health data for a national population and is both sensitive and special category data. Given the detailed nature of the dataset, it is not possible to provide a minimal de-identified dataset that retains the necessary data utility to replicate our study's findings and be considered anonymised. Anonymisation of the Cloudy study data (whilst retaining data utility) is only possible through a combination of measures [i.e., de-identification, data minimisation related to the use case and the provision of access via a Secure Data Environment (SDE)]. These measures are in line with the UK Anonymisation Network guidance (Elliot, Mackey, & O'Hara, 2020). We are currently working towards establishing the processes for supporting access and sharing via an SDE and anticipate the data will be more widely sharable sometime in 2024. Elaine Mackey is the contact for dataset access (elaine.mackey@manchester.ac.uk).

**Funding:** This study was supported by the Centre for Epidemiology Versus Arthritis (grant number 21755). Cloudy with a Chance of Pain was funded by Versus Arthritis (grant reference 21225), with additional support from the Centre for Epidemiology (grants 21755 and 20380). TH receives funding from the Royal Society (grant number INF/R2/180067) and the Alan Turing Institute for Data Science and Artificial Intelligence.

**Competing interests:** I have read the journal's policy and the authors of this manuscript have the following competing interests: WGD has received consultancy fees from Google, and DMS has received consultancy fees from Palta, both unrelated to this work. All other authors have declared that no competing interests exist.

opportunities for different treatment and self-management regimes for different patient subtypes within the four clusters.

## Introduction

Patient-generated health data collected by mobile-health (or mHealth) applications and wearables offer advantages over hand-written diaries or clinical visits (e.g., reviews by [1–5]). Data on exposures and health outcomes can be collected longitudinally, at high frequency, and in patients' own environments. Data can be collected actively (e.g., participants self-reporting pain severity or mood) and passively (e.g., environmental exposures such as the weather). The data enable the assessment of health states that vary over time, the ability to explore covariates associated with health states, and how these associations vary within and between patients.

One health condition that would benefit from patient-generated health data collected via mobile health apps is chronic pain. Chronic pain is common across long-term health conditions, and pain severity is a key outcome measure [6]. Pain severity can vary across different time scales, from minutes and hours to days and beyond. This variability lowers the quality of life of patients, independent of disease severity [7–9], and can increase the risk of depression [10]. More positively, patients with higher pain variability may also respond better to treatment [11]. People living with chronic pain wish to better understand variability in their pain [12], and they report frustration when causes of variability are unknown [13].

Analyzing and understanding daily pain variability is important for (1) allowing patients to self-manage their pain as a step toward anticipating when they might experience a more painful episode, (2) informing clinical care by giving more information to the clinician about the patient's pain and its variability, and (3) providing input to population health research to better understand the causative factors for pain. Importantly, understanding pain variability was ranked of higher importance by patients than by clinicians and researchers from among various pain metrics [14], suggesting a discrepancy between what patients think is important versus what clinicians and researchers think is important. Thus, more importance should be placed on understanding, quantifying, and predicting pain variability, in line with patient preferences.

One approach to understanding pain variability has been to cluster patients by variability in their pain reporting, oftentimes over weeks or months. Different measures of pain variability have been proposed, including standard deviation of the pain variability within an individual (e.g., as reviewed by [15]). However, standard deviation only quantifies the size of the changes, not how the pain changes from day-to-day. A different method that does address the day-to-day variability is clustering. Clustering has been used to understand the course of low back pain over weeks and months [16,17] and in osteoarthritis [18,19], with this course called a *pain trajectory*. Thus, clustering of pain trajectories is a proven method of understanding pain variability.

Previously, [20] identified four clusters of seven-day pain-severity trajectories from daily patient-generated health data. This data came from Cloudy with a Chance of

Pain, a 15-month study involving over 10,000 participants with chronic pain who entered their daily data into a smartphone app [21]. Cloudy with a Chance of Pain was one of the largest to study the relationship between weather and pain (Fig 2 in [22]). [20] used a *k*-medoids algorithm to compare within-week pain severity (but not variability) using the Manhattan distance. Four clusters arose, which were interpretable in plain language as representing "no/low pain", "mild pain", "moderate pain", and "severe pain". Medoid weekly trajectories of these clusters represented pain severity that did not change, but spaghetti plots of trajectories highlighted substantial within-week variability that was not investigated by [20]. This variability may represent important pain-severity dynamics and relationships with exposures that require understanding [23,24].

Therefore, this study investigates the variability within these pain trajectories from [20]. Specifically, the purpose of this article is to (1) quantify the variability in daily pain severity across seven-day periods in the Cloudy with a Chance of Pain dataset within each of the four clusters and (2) identify exposures associated with day-to-day variability in pain severity. In doing so, we aim to understand how pain varies within and among individuals within each of the clusters in the study, perhaps suggesting different patient subtypes that may be associated with different exposures and may benefit from different treatment and self-management regimes.

## Data and methods

In this section, we describe how the data was collected and processed for analysis. We first describe the Cloudy with a Change of Pain study and how the data was collected. We then explain the choice of using seven-day periods and the methodology for calculating the four clusters of weekly pain trajectories. We then describe the quantitative methods for expressing the day-to-day variability within each of the four clusters. Finally, we describe the methods for investigating associations between the day-to-day variability within each of the four clusters and the covariate variables (i.e., patient-generated health-data variables and weather variables).

### Collecting data: The cloudy with a chance of pain study

Data were collected between January 2016 and April 2017 as part of a population-based mobile-health study, Cloudy with a Chance of Pain. Ethical approval for Cloudy with a Chance of Pain was from the University of Manchester Research Ethics Committee (ref: ethics/15522) and the NHS IRAS (ref: 23/NW/0716). The study was described by [21] and subsequently analyzed and reported by [20,25–30]. In brief, people living with long-term pain conditions were recruited through local and national media campaigns [31]. Interested participants downloaded a smartphone app, provided electronic consent, and completed a baseline survey. Baseline data included year of birth, sex (male or female), condition(s) of chronic pain (selected from a list of options, including "other"), and site(s) of chronic pain (selected from a list of options, including "other"). Included participants had at least one chronic-pain condition, were aged ≥ 17 years, were residents of the United Kingdom, and had a smartphone.

Daily data were collected actively and passively via the app (see below for details). Respondents were encouraged to enter their data once a day, in line with anecdotal evidence of daily pain changes due to the time scale of weather changes, as well as not overburdening data entry. Although there is evidence that pain varies within-days [32], our analysis approach is unable to capture variability this frequent. Data were time- and date-stamped and transferred to secure servers. Overall, 10,483 participants downloaded the app, provided full baseline data, and provided at least one report of pain severity. Reporting of this secondary analysis follows the STROBE guidelines [33].

Actively collected patient-generated health data were ten self-reported variables measured using a five-point ordinal scale. The primary outcome was severity of chronic pain. Responses to "How severe was your pain today?", ranged from 1 (no pain) to 5 (very severe pain). Previous studies (as reviewed by [34]) showed a 20% increase in pain is clinically significant, so our five-point scale is appropriate for collection of such patient-generated health data. Other variables were (with value labels for scores 1 and 5 shown): morning stiffness (no stiffness, very severe stiffness), mood (very happy,

depressed), fatigue (no fatigue, very severe fatigue), sleep quality (very good, very poor), tiredness on waking (not at all tired, extremely tired), time spent outside (all of the day, none of the day), exercise (30 + minutes strenuous activity, no exercise), how much their pain interfered with their activities (hereafter, pain interference) (not at all, very much), and well-being (very well, very unwell). Because two persons who report the same level of pain on our scale may experience pain differently, we address daily pain changes within a respondent. Our approach similarly applies to all ten self-reported variables.

Passively collected weather data (i.e., wind speed, temperature, dewpoint temperature, atmospheric pressure, relative humidity) were collected hourly from the nearest weather station identified using the phone's Global Positioning System (GPS; [35]). Mean daily weather conditions were calculated for each participant based on the hourly data.

## Constructing weekly pain-trajectory data

Variability in pain severity was examined across seven days, Monday to Sunday. The choice of seven days was a balance between being long enough to identify a respondent's pain trajectory and short enough that a complete data record without imputing missing data entries for each respondent (i.e., respondents responded every day in a row for seven days) occurs enough to produce a large dataset [20]. Seven days starting on a Monday was also selected because patients often report more pain during weekdays and less pain during weekends, so aligning the period with the workweek accounted for such day-of-the-week effects. The downside of ensuring complete data is that data gaps may not be missing at random (i.e., respondents may not have entered their data during painful periods). Also, there may be differences between complete weeks, and those that were available might have had missing data. For example, people who did not provide complete weeks may have had stable pain and did not perceive a benefit to contributing the same data each day. Previous analyses in identifying clusters of pain severity found no difference between weeks described in this way, as well as weeks defined across other spans (e.g., Tuesday–Monday) [20], but this may not remain true for pain variability.

*Available weeks* were defined as weeks where participants had submitted daily pain severity data, Monday to Sunday. There were 21,919 available weeks contributed by 2807 participants.

*Complete weeks* were defined as available weeks during which complete covariate (for each of the nine patient-generated health-data variables and the weather variables listed above) and lagged covariate data were available. Participants could contribute up to 64 complete weeks (the length of the study). Of the 21,919 available weeks contributed by 2807 participants, 13,052 (59.5%) were classified as complete weeks contributed by 2070 participants. The analyses presented here were based on these 13,052 complete weeks.

For available weeks, the mean age of corresponding participants, the proportion of weeks contributed by females, by individuals with each chronic pain condition, and by individuals with each site of pain were calculated. Sensitivity analyses identified differences in these variables between the available weeks that were and were not included as complete weeks. To test for a difference in the distributions of ages, a Kolmogorov–Smirnoff test was used. To test for differences in the proportions of weeks contributed by females, by people with each chronic pain condition, and by people with each site of pain, chi-squared tests were conducted. When available weeks were incomplete and hence excluded from the analysis, the variable(s) with missing data were recorded.

## Constructing weekly clusters of pain trajectories and quantifying day-to-day variability

In a cluster analysis of patient-generated pain-severity data from Cloudy with a Chance of Pain, [20] reported four clusters representing distinct patterns of weekly pain severity: no/low pain, mild pain, moderate pain, and severe pain. However, not every trajectory within each cluster looked the same. To illustrate different kinds of *daily* variability within a week, Fig 1 shows four examples of trajectories within the moderate-pain cluster, which was characterized by an average pain score of 3 on the scale 1–5. Trajectory 1 shows no day-to-day changes, trajectory 2 shows frequent but moderately sized changes, trajectory 3 shows infrequent but large changes, and trajectory 4 shows large and frequent changes.

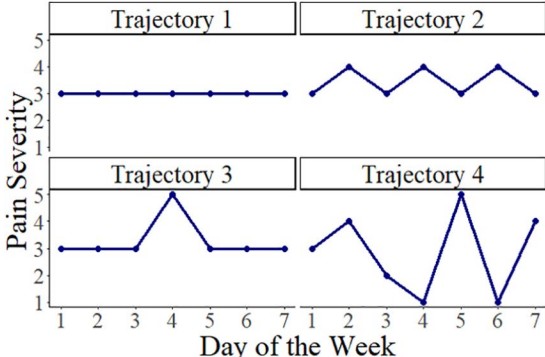

**Fig 1. Four example weekly pain trajectories, taken from data assigned to the moderate-pain cluster.** Trajectory 1 has no variability, trajectory 2 has frequent but small changes, trajectory 3 has infrequent but larger changes, and trajectory 4 has frequent and large changes.

In this study, between-day changes in pain severity were used to quantify variability. A complete week had six between-day changes, taking values between −4 and +4, with 0 being no change. For each pair of consecutive days in available weeks, a *between-day change in pain score* was calculated, as in a hypothetical example in Fig 2, where the participant recorded pain severity of 3 on Sunday, followed by three days at 4, followed by two days at 5, then dropping to 4 on the next Sunday. Similarly, for the same pair of consecutive days, a *between-day change in covariate score* (for each of the nine patient-generated health-data variables and the weather variables listed above) were calculated. For the example in Fig 2, the participant reported a fatigue score of 3 on Sunday, followed by two days of 2, followed by four days of 3, followed by 2 on the next Sunday. For those days with missing data, the between-day change in pain or covariate scores was not calculated. *Lagged between-day changes in covariate scores* were calculated for the previous pair of consecutive days. To calculate lagged variables required covariate data to be completed Sunday to Sunday. To match the structure of the ten self-reported quantities, weather variables were re-scaled with a minimum value of 1 and a maximum value of 5, and between-day changes between weather variables were calculated to range between −4 and +4.

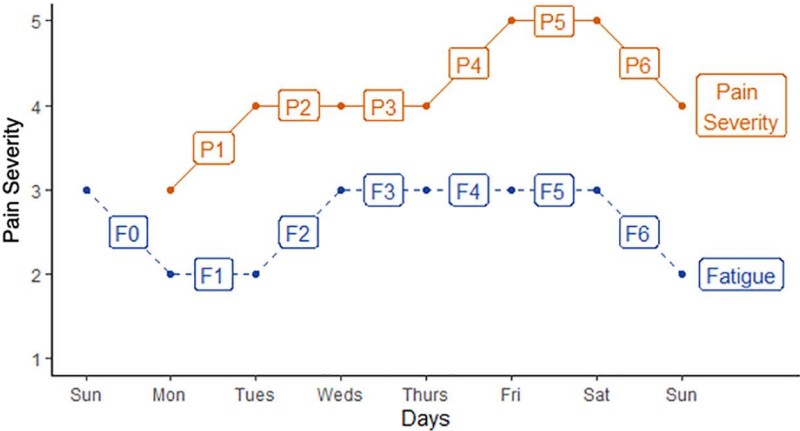

**Fig 2. Pain severity and fatigue scores for a hypothetical participant, with between-day changes for a *complete week* (i.e., where a participant has contributed, in this case, pain severity each day Monday to Sunday and fatigue scores each day Sunday to Sunday).** Dots indicate daily pain scores (in orange) and daily fatigue scores (in blue). P1 denotes the first between-day change in pain scores and has a value of +1, P2 denotes the second between-day change in pain scores and has a score of zero, etc. F1 is a concurrent change in fatigue (with value 0) with P1, and F0 is a lagged change in fatigue (with value −1).

To quantify within-week variability, weeks were categorized as either a *stable-pain week* (all between-day pain changes were zero; e.g., trajectory 1 in Fig 1) or a *variable-pain week* (at least one non-zero between-day pain change; e.g., trajectories 2, 3, and 4 in Fig 1). For variable-pain weeks, the following measures of pain variability were calculated, following [15]:

a) Probability of acute change (PAC). The PAC is the proportion of between-day changes with magnitude ≥ 1.

b) Autocorrelation coefficient (AC) with lag 1. The AC takes continuous values between −1 and 1, with values close to −1 representing a back-and-forth trajectory alternating between being above and below the mean pain severity, and values close to 1 representing trajectories that are similar on consecutive days (e.g., pain above the mean on one day remains above the mean on the following day). Values near zero represent little correlation between values on consecutive days.

c) Mean square of successive difference (MSSD). The MSSD squares the between-day changes in pain severity and then calculates the mean of these values. A higher MSSD is associated with more frequent, or more severe, changes.

d) Proportion of between-day changes observed at each value (−4, −3, −2, −1, 0, +1, +2, +3, +4).

The mean and standard deviation of PAC, AC, and MSSD are reported by cluster, and the proportions of between-day changes are reported as within-cluster distributions.

## Identifying variables associated with day-to-day variability in pain severity

Given our understanding of how pain changes from day to day for participants in various clusters, the next analysis aimed to identify variables that changed before, or concurrently with, changes in pain severity. The method is described as follows.

Here, the outcome variable was defined as the between-day direction of pain change (i.e., negative, positive, zero). Cumulative probit models were used to model this ordinal outcome [36] using the ordinalNet package in R [37]. Models were fit separately for each cluster than within each subgroup of data. This approach avoids masking of associations as per Simpson's Paradox [38], in which opposite associations can be observed at a population level.

There were 50 candidate predictors for each of the four models: baseline variables and concurrent and lagged changes in each of the daily covariate variables. Candidate predictors were standardized by subtracting the mean (for continuous variables) or median (for ordinal variables). Due to the large number of predictors, and the possibility of collinearity among predictors, variable selection was conducted for the probit model using the LASSO [39]. LASSO introduces a shrinkage factor to the likelihood function of the probit model. The strength of this shrinkage factor is controlled by the tuning parameter $\lambda$. The LASSO algorithm shrinks some coefficients to zero, effectively eliminating them from the model. The LASSO is widely used in variable selection [40] due to its ability to identify groups of important variables to be retained in the model and because it is robust to reasonable collinearity between candidate variables.

To select a model with an appropriate reduction of the variables, 20 values of $\lambda$ were tested. For each value of $\lambda$, a probit model was fitted. Of the 20 candidate models, the model with the smallest Bayesian information criterion was selected. This criterion trades off goodness-of-fit against model complexity and has the benefit of increasing the complexity penalty for larger datasets such as ours, providing an extra guard against the risk of overfitting to big data.

Of the optimal model for each cluster, the predictors that were retained in the model (i.e., whose coefficient was not shrunk to zero) are reported. To quantify the baseline probability of each outcome (pain increase, pain decrease, no pain change), all predictors were set to zero in each optimal model. The baseline probabilities of a pain increase and a pain decrease are reported. The marginal probability of each retained predictor was calculated by setting all predictors to zero except the predictor of interest. The marginal probability of each predictor was calculated both by increasing and decreasing the predictor by one unit. For each cluster, the four predictors resulting in the largest changes in probability outcomes are reported graphically. Marginal probabilities for all predictors are reported.

 

# Results

Demographic data is presented in Table 1. The weeks included in the analyses were significantly more likely to be contributed by younger participants; participants who had diagnoses of gout or unspecific arthritis; participants who reported pain in the stomach/abdomen, hip, knee, or hands; and participants who were significantly less likely to have diagnoses of fibromyalgia or to report pain in the head, face, mouth/jaws, or neck/shoulder. The missing variables that result in available weeks being incomplete are reported in Table 2. These differences between the demographic data of available weeks and those that were complete may mean that the observed associations are particularly relevant for the contributors to this analysis, but not generalizable to the wider study population.

Having examined the composition and structure of this dataset, we are now able to address the two-fold purpose of this study from the introduction: quantifying the variability in daily pain severity and identifying exposures associated with this variability.

**Table 1. Demographic data of weeks included as complete weeks and those that were available but not complete. Bold are those that are statistically significant at the 95% level.**

| | | Data of complete weeks | Data of available weeks that were not complete | *p*-value of differences between groups |
|---|---|---|---|---|
| Sex (%) | | | | |
| | Female | 82.7 | 82.1 | 0.237 |
| | Male | 17.3 | 17.9 | 0.237 |
| Age (mean) | | 53.0 | 53.6 | **0.002** |
| Chronic pain condition (%) | | | | |
| | Rheumatoid arthritis | 19.2 | 19.6 | 0.557 |
| | Osteoarthritis | 41.0 | 40.0 | 0.151 |
| | Spondyloarthropathy | 8.6 | 9.0 | 0.243 |
| | Gout | 3.7 | 3.0 | **0.005** |
| | Unspecific arthritis | 43.6 | 40.2 | **<0.001** |
| | Fibromyalgia | 22.8 | 24.2 | **0.019** |
| | Chronic headache | 7.3 | 7.9 | 0.095 |
| | Neuropathic pain | 14.4 | 15.0 | 0.246 |
| | Other/no medical diagnosis | 21.0 | 21.3 | 0.587 |
| Site of pain (%) | | | | |
| | Head | 13.6 | 15.3 | **<0.001** |
| | Face | 4.4 | 5.8 | **<0.001** |
| | Mouth/jaws | 12.1 | 14.1 | **<0.001** |
| | Neck/shoulder | 57.9 | 59.4 | **0.030** |
| | Back | 55.9 | 55.0 | 0.239 |
| | Stomach/abdominal | 13.0 | 11.9 | **0.023** |
| | Hip | 51.7 | 49.1 | **<0.001** |
| | Knee | 66.4 | 63.1 | **<0.001** |
| | Hands | 63.8 | 62.5 | **0.046** |
| | Feet | 47.9 | 47.1 | 0.264 |
| | Multisite | 43.2 | 42.0 | 0.084 |
| | All | 14.6 | 13.8 | 0.111 |

**Table 2. Number (%) of available weeks in which listed variables contained missing data.**

| Fatigue | 4168 (19.0%) |
|---|---|
| Mood | 4158 (19.0%) |
| Morning stiffness | 4640 (21.2%) |
| Pain impact | 4327 (19.7%) |
| Patient wellbeing | 4276 (19.5%) |
| Exercise | 4554 (20.8%) |
| Sleep quality | 4856 (22.2%) |
| Time spent outside | 4679 (21.3%) |
| Waking up tired | 4787 (21.8%) |
| Daily mean windspeed | 1553 (7.1%) |
| Daily mean temperature | 995 (4.5%) |
| Daily mean dewpoint | 995 (4.5%) |
| Daily mean pressure | 995 (4.5%) |
| Daily mean relative humidity | 995 (4.5%) |

## Quantifying day-to-day variability in pain severity among the four clusters

The proportion of available weeks classified as complete weeks was similar across clusters: the no/low-pain cluster contained 56.6% ($n=970$) complete weeks, the mild-pain cluster contained 59.2% ($n=4885$), the moderate-pain cluster contained 60.1% ($n=5036$), and the severe-pain cluster contained 60.3% ($n=2161$). The 2070 participants provided an average of 6.3 complete weeks, and 95% of the participants contributed between 1 and 28 weeks.

Available weeks were incomplete if they contained missing data for at least one of the listed variables. The number (%) of available weeks with missing data is reported. Available weeks may have contained missing data for multiple variables, and therefore weeks may be counted multiple times. To quantify the variability, Table 3 reports the number and percentage of complete weeks that were stable-pain weeks by cluster. In total, 1474 (11.3%) complete weeks were stable. A higher percentage of weeks were stable in the no/low-pain cluster (21.9%) when compared to the other three clusters. The moderate-pain cluster contains the lowest percentage of stable-pain weeks (7.4%).

Among variable-pain weeks, Fig 3 illustrates the percentage of pain differences observed by cluster. Complete weeks in the no/low-pain cluster showed the lowest percentage of changes (59% of consecutive days had no pain change), whereas the moderate-pain cluster showed the highest percentage of pain changes (48% of consecutive days had no pain change). When pain did change, the change was most likely by an absolute difference of 1 unit, but at least 6% of observed pain differences had a magnitude of ≥ 2 units. Overall, 0.06% of pain changes were differences of –4 or +4 and were more likely in the no/low-pain cluster and the severe-pain cluster, likely due to pain severity in these clusters being at extreme ends of the scale and therefore permitting greater movement.

Table 4 quantifies the measures of variability among variable-pain weeks. The PAC showed that, on average, trajectories in the no/low-pain cluster changed less frequently (0.41) than clusters of more severe pain (e.g., moderate-pain cluster: 0.52). The average AC (with lag 1) was close to zero in each cluster, suggesting little correlation between days on average. On any individual week, however, the AC values range from –0.86 to +0.67, with an interquartile range of –0.26 to +0.23. Finally, the MSSD values were greater in clusters with more severe pain, indicating more between-day variability in these clusters. Some of this greater variability is captured by the PAC, but it is unclear from these summary measures alone whether one cluster has more severe changes.

**Table 3. Percentage of stable trajectories.**

| Cluster | Number (percentage) of complete stable-pain weeks | Description of complete stable-pain weeks |
|---------|---------------------------------------------------|-------------------------------------------|
| no/low pain | 212 (21.9%) | All are constant '1' pain |
| mild pain | 647 (13.2%) | All are constant '2' pain |
| moderate pain | 374 (7.4%) | All are constant '3' pain |
| severe pain | 241 (11.2%) | 94 (4.3% of total) are constant '4' pain, and 147 (6.8% of total) are constant '5' pain |

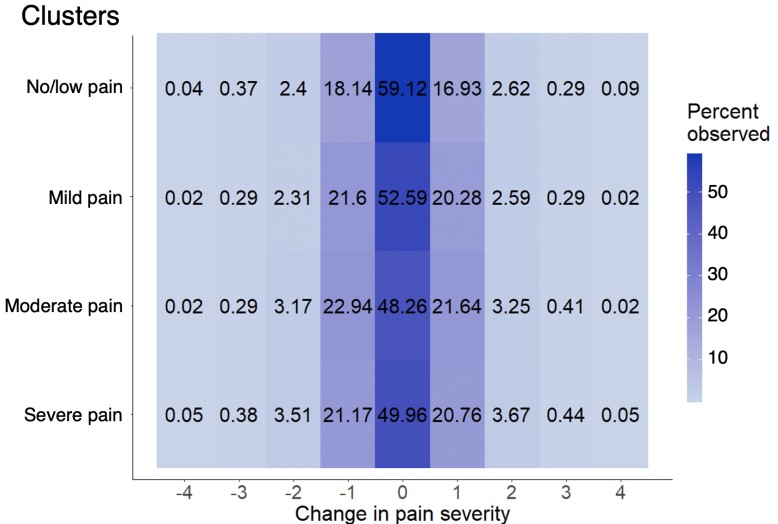

**Fig 3. Distribution of pain changes within variable-pain weeks.**

**Table 4. Measures of variability by cluster.**

| Cluster | PAC among variable trajectories: mean (SD) | AC among variable trajectories: mean (SD) | MSSD among variable trajectories: mean (SD) |
|---------|---------------------------------------------|--------------------------------------------|---------------------------------------------|
| No/low pain ($n=680$) | 0.409 (0.191) | –0.002 (0.300) | 0.63 (0.61) |
| Mild pain ($n=3788$) | 0.474 (0.213) | –0.049 (0.296) | 0.67 (0.58) |
| Moderate pain ($n=4155$) | 0.517 (0.215) | –0.036 (0.302) | 0.77 (0.66) |
| Severe pain ($n=1745$) | 0.500 (0.214) | –0.019 (0.304) | 0.80 (0.73) |

## Variables associated with day-to-day variability in pain severity among the four clusters

Next, we aim to identify the variables associated with pain variability. For each optimal model identified, and given no change in covariates, the baseline probabilities of experiencing an increase or decrease in pain severity are reported in Table 5 and visualized by horizontal lines in Fig 4. With no changes in predictor variables, day-to-day pain changes had a probability of improvement (i.e., positive values of between-day changes) of 0.128 in the no/low-pain cluster, 0.172 in the mild-pain cluster, 0.202 in the moderate-pain cluster, and 0.182 in the severe-pain cluster, and a probability of worsening

**Table 5. Probability of pain recovery and pain worsening, at baseline, and given one-unit changes in significant variables, by cluster.**

| | No/low-pain cluster | | Mild-pain cluster | | Moderate-pain cluster | | Severe-pain cluster | |
|---|---|---|---|---|---|---|---|---|
| | P(Pain recovery) | P(Pain worsening) | P(Pain recovery) | P(Pain worsening) | P(Pain recovery) | P(Pain worsening) | P(Pain recovery) | P(Pain worsening) |
| Baseline | 0.128 | 0.120 | 0.172 | 0.163 | 0.202 | 0.192 | 0.182 | 0.180 |
| Worsened fatigue by 1 unit | 0.117 | 0.131 | 0.151 | 0.186 | 0.173 | 0.223 | 0.135 | 0.235 |
| Improved fatigue by 1 unit | 0.139 | 0.109 | 0.196 | 0.142 | 0.235 | 0.163 | 0.237 | 0.134 |
| Worsened mood by 1 unit | 0.122 | 0.126 | 0.164 | 0.172 | 0.187 | 0.207 | 0.167 | 0.195 |
| Improved mood by 1 unit | 0.134 | 0.114 | 0.181 | 0.155 | 0.218 | 0.177 | 0.197 | 0.166 |
| More exercise by 1 unit | . | . | 0.165 | 0.171 | 0.193 | 0.201 | 0.172 | 0.191 |
| Less exercise by 1 unit | . | . | 0.180 | 0.156 | 0.212 | 0.183 | 0.192 | 0.171 |
| Reduced time spent outside by 1 unit | . | . | . | . | 0.211 | 0.183 | 0.192 | 0.171 |
| Increased time spent outside by 1 unit | . | . | . | . | 0.193 | 0.200 | 0.172 | 0.190 |
| Worsened stiffness by 1 unit | 0.063 | 0.217 | 0.112 | 0.239 | 0.139 | 0.267 | 0.128 | 0.246 |
| Improved stiffness by 1 unit | 0.229 | 0.058 | 0.250 | 0.105 | 0.280 | 0.131 | 0.248 | 0.127 |
| Worsened lag stiffness by 1 unit | . | . | 0.167 | 0.169 | . | . | . | . |
| Improved lag stiffness by 1 unit | . | . | 0.178 | 0.158 | . | . | . | . |
| Worsened sleep by 1 unit | 0.122 | 0.126 | 0.165 | 0.170 | 0.196 | 0.198 | 0.179 | 0.183 |
| Improved sleep by 1 unit | 0.134 | 0.114 | 0.180 | 0.156 | 0.209 | 0.185 | 0.185 | 0.177 |
| Worsened pain interference by 1 unit | 0.020 | 0.395 | 0.065 | 0.340 | 0.097 | 0.343 | 0.097 | 0.301 |
| Improved pain interference by 1 unit | 0.411 | 0.018 | 0.354 | 0.061 | 0.357 | 0.090 | 0.303 | 0.096 |
| Worsened lag pain interference by 1 unit | . | . | 0.167 | 0.169 | 0.192 | 0.202 | . | . |
| Improved lag pain interference by 1 unit | . | . | 0.178 | 0.158 | 0.213 | 0.181 | . | . |
| Worsened wellbeing by 1 unit | 0.110 | 0.139 | 0.142 | 0.196 | 0.165 | 0.232 | 0.149 | 0.218 |
| Improved wellbeing by 1 unit | 0.148 | 0.102 | 0.207 | 0.134 | 0.244 | 0.156 | 0.219 | 0.148 |
| Increased lag windspeed by 1 unit | . | . | . | . | 0.208 | 0.186 | . | . |
| Decreased lag windspeed by 1 unit | . | . | . | . | 0.197 | 0.197 | . | . |

(i.e., negative values of between-day changes) of 0.120 in the no/low-pain cluster, 0.163 in the mild-pain cluster, 0.192 in the moderate-pain cluster, and 0.180 in the severe-pain cluster.

To determine the likelihood of pain worsening or improving, Table 5 shows the marginal probability of pain worsening or improving given exposure data with non-zero coefficients for the four clusters. Fig 4 shows the four variables with the largest changes in marginal probabilities: changes in pain interference, morning stiffness, well-being, and fatigue. A one-unit change in pain interference had the greatest impact on the probability of a change in pain severity, held true across all clusters. Increases in pain interference were associated with increases in the probability of day-to-day pain worsening to 0.395 in the no/low-pain cluster (up by 0.275 from baseline), 0.340 in the mild-pain cluster (up by 0.177 from baseline), 0.343 in the moderate-pain cluster (up by 0.151 from baseline), and 0.301 in the severe-pain cluster (up by 0.121 from baseline). Increases in pain interference were also associated with decreases in the probability of day-to-day pain improvement to 0.020 in the no/low-pain cluster (down by 0.108 from baseline), 0.065 in the mild-pain cluster (down by 0.107 from baseline), 0.097 in the moderate-pain cluster (down by 0.105 from baseline), and 0.097 in the severe-pain cluster (down by 0.085 from baseline). Thus, the LASSO method is generally robust for improving model performance in the presence of collinear predictors but the increased suitability of selected covariates over deselected collinear covariates is unclear.

The LASSO variable selection occurred with different levels of shrinkage ($\lambda$). Then, fatigue, mood, stiffness upon waking, sleep quality, pain interference, and patient well-being were retained in all models, indicating associations between

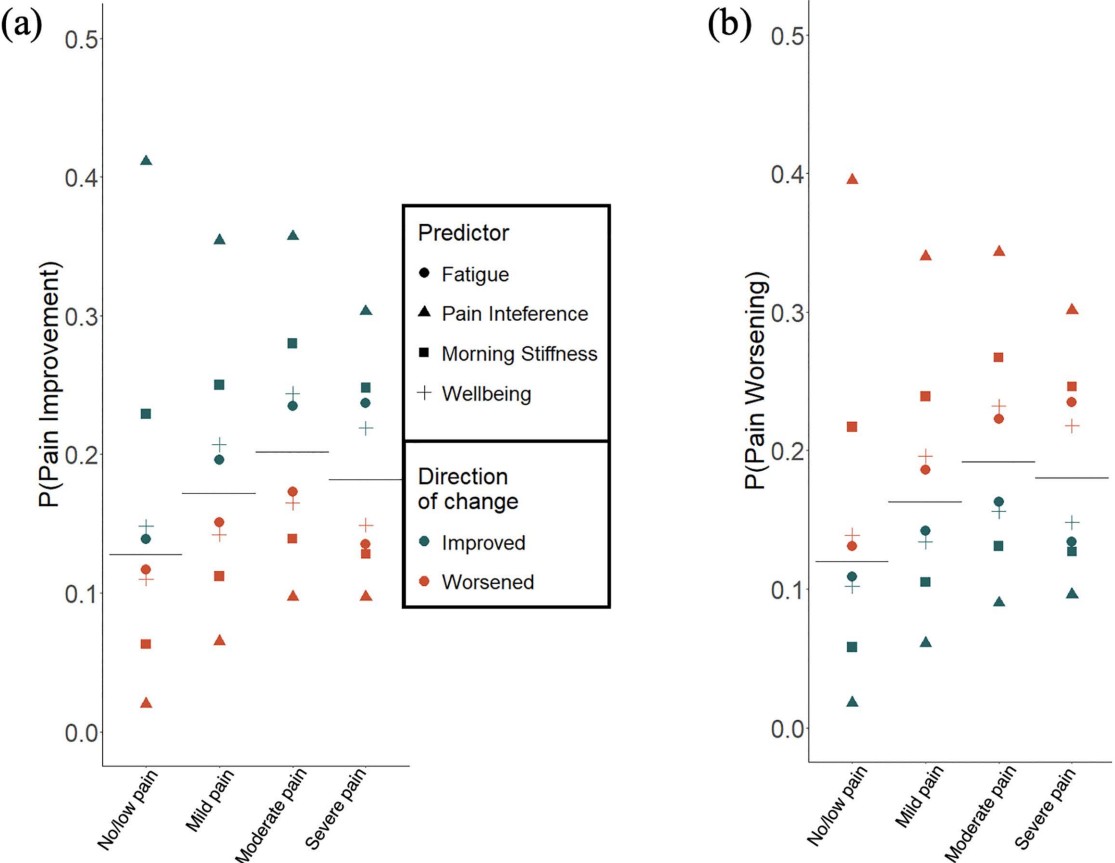

**Fig 4. (a) Baseline (no change in patient-generated health and weather variables) probability of pain severity improving shown by horizontal line for each cluster. (b) Baseline probability of pain severity worsening shown by horizontal line for each cluster.** Points represent probability of pain severity improving given an improvement of one-unit (green) or worsening of one-unit (red) of fatigue, pain interference, morning stiffness, and wellbeing.

concurrent changes in these variables and in pain severity across all clusters of pain severity (Table 6). Some evidence of cluster-specific associations can be seen. For example, pain changes were associated with lag morning stiffness in the mild-pain cluster only, with exercise in all the clusters except no/low pain, with lag wind speed in the moderate-pain cluster, and with time spent outside in the moderate- and severe-pain clusters (Table 6).

## Discussion

Cloudy with a Chance of Pain was a mobile-health study that collected daily patient-generated health and weather data via a smartphone app. Daily data were used to identify the weekly trajectories of pain severity among the participants. Pain severity, and other data were recorded on a scale from 1 (no pain) to 5 (severe pain). A previously published clustering of the data revealed four clusters representing no/low pain, mild pain, moderate pain, and severe pain. Within each cluster, individual trajectories could show substantial within-week variability in pain severity. The between-day change in pain was therefore indicated by a number in the range from –4 to +4. This article aimed to quantify this variability across the four clusters and identify the variables associated with this variability.

We identified stable-pain weeks where no day-to-day change in pain occurred and variable-pain weeks where at least one change was nonzero. Within each cluster, variable-pain weeks had no pain changes in 48–59% of consecutive days,

**Table 6. Variables associated with changes in pain.** Variables retained by models following shrinkage through LASSO are shown by the blue coloring. Changes are concurrent with pain change unless otherwise stated.

| Cluster | No/low pain | Mild pain | Moderate pain | Severe pain |
|---|---|---|---|---|
| Fatigue | ■ | ■ | ■ | ■ |
| Mood | ■ | ■ | ■ | ■ |
| Exercise | | ■ | ■ | ■ |
| Time spent outside | | | ■ | ■ |
| Morning stiffness | ■ | ■ | ■ | ■ |
| Lag morning stiffness | | ■ | | |
| Sleep quality | ■ | ■ | ■ | ■ |
| Pain interference | ■ | ■ | ■ | ■ |
| Lag pain interference | | | ■ | ■ |
| Well-being | ■ | ■ | ■ | ■ |
| Lag wind speed | | | ■ | |

with the no/low-pain cluster showing the lowest percentage of changes (59%) and the moderate-pain cluster showing the highest (48%). Non-zero changes were most likely to be one unit in magnitude, with 6% being ≥ 2 units.

Across all clusters, the PAC indicated pain changes on around half of the observed days. The AC was close to zero, but with a large range and interquartile range; therefore, no population-level conclusion could be drawn from this measure. The MSSD, taken alongside the distribution of pain changes, indicated that moderate- and severe-pain clusters exhibited more severe changes. However, the no/low- and severe-pain clusters had the highest proportion of extreme changes (−4 or +4), likely due to the opportunity for greater movement between pain severity scores in these clusters.

These results suggest that if pain severity is low over a week (no/low-pain cluster), then it is more likely to be stable, whereas weeks with moderate or severe pain (moderate- and severe-pain clusters) are more likely to have day-to-day changes. Across all clusters, changes in pain interference, well-being, morning stiffness, fatigue, sleep quality, and mood were associated with changes in pain severity. For other predictors, there was evidence of cluster-specific associations (e.g., pain changes were associated with lag morning stiffness in the mild-pain cluster only, with exercise in all the clusters except no/low pain, with lag wind speed in the moderate-pain cluster, and with time spent outside in the moderate- and severe-pain clusters).

Our previous work [20] measured pain variability by using a single statistic, most commonly the within-trajectory standard deviation. This statistic summarizes the magnitude of distance from the mean but has no temporal features and so does not measure day-to-day changes. [15] described three other measures of pain variability (PAC, AC, and MSSD), and we have explored these in this study. Each of these measures has drawbacks for understanding pain variability. PAC requires a subjective cut-off point. AC summarizes temporal characteristics, but not the magnitude, of changes. MSSD is difficult to interpret for participants and clinicians. Summarizing all of these, as well as distributions of pain variability across the clusters, has provided a broader description of pain variability than could be afforded by any single measure.

Our results are consistent with other clusters found in the literature on pain trajectories [16,17]. Specifically, [16] found four clusters in a study of low back pain, two of which are similar to the low/no-pain and severe-pain clusters in our study ("persistent mild" and "severe chronic", respectively). A third ("fluctuating") may be similar to our moderate-pain cluster, categories also synthesized from a range of studies by [17].

As to the associations with variability, similar to our study, [41] reported that low pain severity was associated with low pain variability. Although associations between pain variability and exposure data (mental and physical health and functioning) have been previously reported [42], some studies [10,43] found no predictors of pain variability. In contrast, our

study found that, in all clusters of pain severity, pain changes were associated with changes in pain interference, morning stiffness, patient well-being, and fatigue.

This study has several strengths. We described pain variability using multiple measures in a large dataset of daily pain-severity data (2070 participants lasting over 15 months). Avoiding the use of a single summary statistic provided a more holistic description of pain variability. We used previously identified clusters to identify differences in pain variability among these clusters (in this case, based on the severity of pain). Exploring variability at a group-level avoided masking of population-level associations and identified cluster-specific associations. Finally, we tested associations of pain change with both concurrent and lagged variables, including passively collected data.

This study also possessed several limitations. First, relying on self-reporting data led to a substantial number of weeks of data that were omitted due to incomplete covariate data. We don't know whether the omitted data from incomplete weeks would differ systematically in key outcomes (pain levels, variability) compared to the data that was included from complete weeks. Second, the demographics of the study were highly skewed toward females and toward younger patients. Preliminary indications are that the demographics do not affect the results significantly, but further research is required to be confident.

People with chronic pain want to know about future changes in pain severity [12]. We have shown that daily collected patient-generated health data can be used to summarize pain variability and associated predictors of that variability. The methods for summarizing pain variability in this study are not specific to musculoskeletal chronic pain and could be used in other fields (e.g., migraines, mental health) to explore variability in daily symptoms. Future work can use the identified associations to explore the feasibility of forecasting future changes in pain severity. Some of the identified associations are modifiable (e.g., mood can be improved with psychological support) and could be the focus of future work to reduce pain variability.

## Acknowledgments

We thank Ridvan Isik and an anonymous reviewer for comments that improved this article.

## Author contributions

**Conceptualization:** Claire L. Little, Belay B. Yimer, Thomas House, William G. Dixon, David M. Schultz, John McBeth.

**Formal analysis:** Claire L. Little.

**Funding acquisition:** William G. Dixon.

**Investigation:** Claire L. Little.

**Methodology:** Claire L. Little, Belay B. Yimer.

**Project administration:** William G. Dixon.

**Supervision:** Thomas House, William G. Dixon, David M. Schultz, John McBeth.

**Visualization:** Claire L. Little.

**Writing – original draft:** Claire L. Little.

**Writing – review & editing:** Claire L. Little, Belay B. Yimer, Thomas House, William G. Dixon, David M. Schultz, John McBeth.

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
