## [Decision Letter · Decision Letter 0]

9 Dec 2025

PONE-D-25-45216Patterns and predictors of variability in patient-generated daily pain severity collected via a mobile health smartphone appPLOS One

Dear Dr. Schultz,

Thank you for submitting your manuscript to PLOS ONE. After careful consideration, we feel that it has merit but does not fully meet PLOS ONE’s publication criteria as it currently stands. Therefore, we invite you to submit a revised version of the manuscript that addresses the points raised during the review process.

We look forward to receiving your revised manuscript.

Kind regards,

Mohamad K. Abou Chaar, M.D.

Academic Editor

PLOS One

Journal Requirements:

When you resubmit, please ensure that you provide the correct details and grant numbers for the awards you received for your study in the ‘Funding Information’ section.

[This study was supported by the Centre for Epidemiology Versus Arthritis (grant number 21755). Cloudy with a Chance of Pain was funded by Versus Arthritis (grant reference 21225), with additional support from the Centre for Epidemiology (grants 21755 and 20380). TH receives funding from the Royal Society (grant number INF/R2/180067) and the Alan Turing Institute for Data Science and Artificial Intelligence.].

[This study was supported by the Centre for Epidemiology Versus Arthritis (grant number 21755). Cloudy with a Chance of Pain was funded by Versus Arthritis (grant reference 21225), with additional support from the Centre for Epidemiology (grants 21755 and 20380). Ethical approval for Cloudy with a Chance of Pain was from the University of Manchester Research Ethics Committee (ref: ethics/15522) and the NHS IRAS (ref: 23/NW/0716). TH receives funding from the Royal Society (grant number INF/R2/180067) and the Alan Turing Institute for Data Science and Artificial Intelligence. The funders of the study had no role in study design, data interpretation, writing of the manuscript, or the decision to submit for publication.]

[This study was supported by the Centre for Epidemiology Versus Arthritis (grant number 21755). Cloudy with a Chance of Pain was funded by Versus Arthritis (grant reference 21225), with additional support from the Centre for Epidemiology (grants 21755 and 20380). TH receives funding from the Royal Society (grant number INF/R2/180067) and the Alan Turing Institute for Data Science and Artificial Intelligence.]

[I have read the journal's policy and the authors of this manuscript have the following competing interests: WGD has received consultancy fees from Google, and DMS has received consultancy fees from Palta, both unrelated to this work. All other authors have declared that no competing interests exist.].

6. You have indicated that data is available from [elaine.mackey@manchester.ac.uk].  Please can we ask you to provide us with a general contact email address for the data requests, so readers can request access in perpetuity. If a general email is not available please provide a link to a website where readers can obtain access to data.

7. Your ethics statement should only appear in the Methods section of your manuscript. If your ethics statement is written in any section besides the Methods, please move it to the Methods section and delete it from any other section. Please ensure that your ethics statement is included in your manuscript, as the ethics statement entered into the online submission form will not be published alongside your manuscript.

Reviewers' comments:

Reviewer's Responses to Questions

**Comments to the Author**

1. Is the manuscript technically sound, and do the data support the conclusions?

Reviewer #1: Yes

Reviewer #2: Yes

2. Has the statistical analysis been performed appropriately and rigorously? 

Reviewer #1: I Don't Know

Reviewer #2: Yes

3. Have the authors made all data underlying the findings in their manuscript fully available?

Reviewer #1: Yes

Reviewer #2: Yes

4. Is the manuscript presented in an intelligible fashion and written in standard English?

Reviewer #1: Yes

Reviewer #2: Yes

5. Review Comments to the Author

Reviewer #1: Review of PONE-D-25-45216:

Patterns and predictors of variability in patient-generated daily pain severity collected via a mobile health smartphone app (Schultz et al.)

Overview:

This manuscript addresses an important question: how daily pain variability can be characterized using high-frequency patient-generated data from a large mHealth cohort. The dataset is rich and the basic scientific-clinical question is solid. The work builds on the prior “Cloudy with a Chance of Pain” studies and extends them meaningfully by focusing specifically on variability. That said, this paper has some methodological and mostly reporting flaws that warrant correction before publication.

Major Comments:

1. This paper is extremely technical. As a physician with only moderate understanding of statistics I found it very hard to understand. Moreover, while I have faith in the authors that it is is mathematically correct (too technical for me to review), I could not make out the narrative of the paper, i.e, there is simple “big picture” section guiding the reader through the logic of the analysis before the numbers appear. Key technical terms (AC, PAC, MSSD) are not well explained for the non-expert readers. Understanding these is key to understanding the paper entirely.

2. Similarly, I feel like the paper focuses too much on mathematical findings and too little on patient-centered outcomes. It jumps between terms like stable weeks, variability metrics, LASSO-selected predictors and without interperting these into outcomes that matter to patients. This also menifests in the fact the discussion is very short compared to other parts of the manuscript.

3. The Methods section is in my view overly long and somewhat cumbersome. Honestly, although reading it twice, I am not sure I understood the Methods correctly. It includes many internal definitions (e.g. available weeks, covariate score), the reasoning behing choosing different options in the methods (e.g. why a week starts on Monday), very in-depth statistical data, and goes into so much detail about the methodology that for the reader unfamiliar with the previous work, it becomes impossible to understand. I recommend revising the Methods to make it much shorter – e.g. one parageaph for each of the subsections. I also recommend adding an explanatory figure. If the authors still want to include what’s currently described in the Methods, I recommend adding a supplementary appendix.

4. All tables and figures require legends (which include explanation of acronyms), and should be referenced from and support the arguments made in the main text. Currently many just report a lot of data. Similarly I feel that some data reported in the supplement (e.g. demographics) should be transferred to the main manuscript file, and vice versa.

5. This study includes several limitations that should either be mitigated, better reported, or at the very least mentioned in the Discussion (which oddly does not include a “limitations” section). These include:

A. substantial loss of weeks due to incomplete covariate data

B. Demographics are highly skewed toward females and toward younger patients.

C. It is unclear whether complete weeks differ systematically in key outcomes (pain levels, variability) compared to incomplete weeks.

6. Conclusions section is very long and includes parts that don’t belong in it. Conclusions should be one paragraph summarizing the paper, everything else belongs in the general Discussion.

Minor Comments:

1. Abstract mentions four clusters but does not explain clustering methodology. Suggest including a short explanation as exists in the full paper (e.g. no, mild, moderate and severe pain). In general, for the sake of simplicity I recommend using the actual term (mild-moderate-severe) instead of the A-B-C-D labeling which may add confusion.

2. I find it odd that two figures in the paper describe a hypothetical patient. Figures should depict data collected.

3. References: Schneider et al. (2012) is duplicated.

4. Figure 3b (graphical distribution of pain differences) is informative but difficult to interpret without a color scale or more explicit labeling.

5. Note some minor typos throughout the manuscript (e.g. inteference).

Reviewer #2: I congratulate you for your detailed analysis and meticulously prepared research, as well as for your clearly written article. However, there are some points that I would like to draw attention to. -What does "pain interfence" mean, which you mentioned on page 11, lines 358-359? Could you explain this a little more?

- The conclusion section is longer than the discussion section. Shouldn't the information in the conclusion section be included in the discussion section?

6. PLOS authors have the option to publish the peer review history of their article (what does this mean? ). If published, this will include your full peer review and any attached files.). If published, this will include your full peer review and any attached files.

**Do you want your identity to be public for this peer review?** For information about this choice, including consent withdrawal, please see our For information about this choice, including consent withdrawal, please see our Privacy Policy ..

Reviewer #1: No

Reviewer #2: **Yes:** Ridvan IsikRidvan Isik

---

## [Author Response · Author response to Decision Letter 1]

21 Jan 2026

See attached document Response to Reviewers.docx

---

## [Decision Letter · Decision Letter 1]

25 Feb 2026

PONE-D-25-45216R1Patterns and predictors of variability in patient-generated daily pain severity collected via a mobile health smartphone appPLOS One

Dear Dr. Schultz,

Thank you for submitting your manuscript to PLOS ONE. After careful consideration, we feel that it has merit but does not fully meet PLOS ONE’s publication criteria as it currently stands. Therefore, we invite you to submit a revised version of the manuscript that addresses the points raised during the review process.

Please consider modifying the conclusion. It should be brief and concise, describing only key takeaways from your research. ==============================

We look forward to receiving your revised manuscript.

Kind regards,

Mohamad K. Abou Chaar

Academic Editor

PLOS One

Journal Requirements:

Additional Editor Comments :

Dear Authors,

Thank you for the revision and for addressing all of the comments. I do have one additional comment regarding the conclusion section. To me, the first three paragraphs of the conclusion would fit better in the discussion section. I understand the PLOS ONE format requirements, but at the end of the day, the conclusion should be a brief and focused summary of the main takeaway, its importance, and the key implication of the study rather than an extended discussion. This will add additional overall clarity to your manuscript.

Thank you,

Mohamad K. Abou Chaar, M.D.

Reviewers' comments:

Reviewer's Responses to Questions

**Comments to the Author**

1. If the authors have adequately addressed your comments raised in a previous round of review and you feel that this manuscript is now acceptable for publication, you may indicate that here to bypass the “Comments to the Author” section, enter your conflict of interest statement in the “Confidential to Editor” section, and submit your "Accept" recommendation.

Reviewer #1: (No Response)

Reviewer #2: All comments have been addressed

2. Is the manuscript technically sound, and do the data support the conclusions?

Reviewer #1: Yes

Reviewer #2: Yes

3. Has the statistical analysis been performed appropriately and rigorously? 

Reviewer #1: I Don't Know

Reviewer #2: I Don't Know

4. Have the authors made all data underlying the findings in their manuscript fully available?

Reviewer #1: Yes

Reviewer #2: Yes

5. Is the manuscript presented in an intelligible fashion and written in standard English?

Reviewer #1: Yes

Reviewer #2: Yes

6. Review Comments to the Author

Reviewer #1: I appreciate the authors' work to revise the paper, it is indeed more readable now and with clearly stated findings.

I still think it is overly technical, that some methods and results belong in a supplementary appendix, and that the conclusions section should be further refined, all as I mentioned in my initial review. I understand the authors disagree with these recommendations, and leave the final decision to the authors and the editor.

Reviewer #2: I appreciate your efforts and the time you spent, as you have adequately addressed all comments and suggestions.

7. PLOS authors have the option to publish the peer review history of their article (what does this mean? ). If published, this will include your full peer review and any attached files.). If published, this will include your full peer review and any attached files.

**Do you want your identity to be public for this peer review?** For information about this choice, including consent withdrawal, please see our For information about this choice, including consent withdrawal, please see our Privacy Policy ..

Reviewer #1: No

Reviewer #2: **Yes:** RIDVAN ISIKRIDVAN ISIK

---

## [Author Response · Author response to Decision Letter 2]

26 Feb 2026

See the Responses to Reviewers document.

---

## [Editor Report · Decision Letter 2]

6 Mar 2026

Patterns and predictors of variability in patient-generated daily pain severity collected via a mobile health smartphone app

PONE-D-25-45216R2

Dear Dr. Schultz,

We’re pleased to inform you that your manuscript has been judged scientifically suitable for publication and will be formally accepted for publication once it meets all outstanding technical requirements.

Kind regards,

Mohamad K. Abou Chaar, M.D.

Academic Editor

PLOS One

---

## [Editor Report · Acceptance letter]

PONE-D-25-45216R2

PLOS One

Dear Dr. Schultz,

I'm pleased to inform you that your manuscript has been deemed suitable for publication in PLOS One. Congratulations! Your manuscript is now being handed over to our production team.

Kind regards,

on behalf of

Dr. Mohamad K. Abou Chaar

Academic Editor

PLOS One